# DropEdge: Towards Deep Graph Convolutional Networks on Node Classification

**Yu Rong[1], Wenbing Huang[2]\*, Tingyang Xu[1], Junzhou Huang[1]**
[1] Tencent AI Lab

[2] Beijing National Research Center for Information Science and Technology (BNRist),
State Key Lab on Intelligent Technology and Systems,
Department of Computer Science and Technology, Tsinghua University
`yu.rong@hotmail.com, hwenbing@126.com`
`tingyangxu@tencent.com, jzhuang@uta.edu`

## Abstract

*Over-fitting* and *over-smoothing* are two main obstacles of developing deep Graph Convolutional Networks (GCNs) for node classification. In particular, over-fitting weakens the generalization ability on small dataset, while over-smoothing impedes model training by isolating output representations from the input features with the increase in network depth. This paper proposes DropEdge, a novel and flexible technique to alleviate both issues. At its core, DropEdge randomly removes a certain number of edges from the input graph at each training epoch, acting like a data augmenter and also a message passing reducer. Furthermore, we theoretically demonstrate that DropEdge either reduces the convergence speed of over-smoothing or relieves the information loss caused by it. More importantly, our DropEdge is a general skill that can be equipped with many other backbone models (*e.g.* GCN, ResGCN, GraphSAGE, and JKNet) for enhanced performance. Extensive experiments on several benchmarks verify that DropEdge consistently improves the performance on a variety of both shallow and deep GCNs. The effect of DropEdge on preventing over-smoothing is empirically visualized and validated as well. Codes are released on `https://github.com/DropEdge/DropEdge`.

## 1 Introduction

Graph Convolutional Networks (GCNs), which exploit message passing or equivalently certain neighborhood aggregation function to extract high-level features from a node as well as its neighborhoods, have boosted the state-of-the-arts for a variety of tasks on graphs, such as node classification (Bhagat et al., 2011; Zhang et al., 2018), social recommendation (Freeman, 2000; Perozzi et al., 2014), and link prediction (Liben-Nowell & Kleinberg, 2007) to name some. In other words, GCNs have been becoming one of the most crucial tools for graph representation learning. Yet, when we revisit typical GCNs on node classification (Kipf & Welling, 2017), they are usually shallow (*e.g.* the number of the layers is $2$[1]). Inspired from the success of deep CNNs on image classification, several attempts have been proposed to explore how to build deep GCNs towards node classification (Kipf & Welling, 2017; Li et al., 2018a; Xu et al., 2018a; Li et al., 2019); nevertheless, none of them delivers sufficiently expressive architecture. The motivation of this paper is to analyze the very factors that impede deeper GCNs to perform promisingly, and develop method to address them.

We begin by investigating two factors: *over-fitting* and *over-smoothing*. Over-fitting comes from the case when we utilize an over-parametric model to fit a distribution with limited training data, where the model we learn fits the training data very well but generalizes poorly to the testing data. It does exist if we apply a deep GCN on small graphs (see 4-layer GCN on Cora in Figure 1). Over-smoothing, towards the other extreme, makes training a very deep GCN difficult. As first introduced by Li et al. (2018a) and further explained in Wu et al. (2019); Xu et al. (2018a); Klicpera et al. (2019), graph convolutions essentially push representations of adjacent nodes mixed with each

---

\*Wenbing Huang is the corresponding author.

[1]When counting the number of layers (or network depth) of GCN, this paper does not involve the input layer.

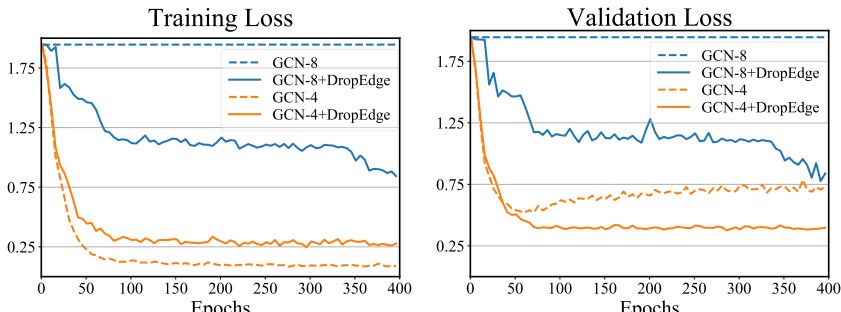

Figure 1: Performance of Multi-layer GCNs on Cora. We implement 4-layer GCN w and w/o DropEdge (in orange), 8-layer GCN w and w/o DropEdge (in blue)[2]. GCN-4 gets stuck in the over-fitting issue attaining low training error but high validation error; the training of GCN-8 fails to converge satisfactorily due to over-smoothing. By applying DropEdge, both GCN-4 and GCN-8 work well for both training and validation.

other, such that, if extremely we go with an infinite number of layers, all nodes' representations will converge to a stationary point, making them unrelated to the input features and leading to vanishing gradients. We call this phenomenon as over-smoothing of node features. To illustrate its influence, we have conducted an example experiment with 8-layer GCN in Figure 1, in which the training of such a deep GCN is observed to converge poorly.

Both of the above two issues can be alleviated, using the proposed method, DropEdge. The term "DropEdge" refers to randomly dropping out certain rate of edges of the input graph for each training time. There are several benefits in applying DropEdge for the GCN training (see the experimental improvements by DropEdge in Figure 1). First, DropEdge can be considered as a data augmentation technique. By DropEdge, we are actually generating different random deformed copies of the original graph; as such, we augment the randomness and the diversity of the input data, thus better capable of preventing over-fitting. Second, DropEdge can also be treated as a message passing reducer. In GCNs, the message passing between adjacent nodes is conducted along edge paths. Removing certain edges is making node connections more sparse, and hence avoiding over-smoothing to some extent when GCN goes very deep. Indeed, as we will draw theoretically in this paper, DropEdge either reduces the convergence speed of over-smoothing or relieves the information loss caused by it.

We are also aware that the dense connections employed by JKNet (Xu et al., 2018a) are another kind of tools that can potentially prevent over-smoothing. In its formulation, JKNet densely connects each hidden layer to the top one, hence the feature mappings in lower layers that are hardly affected by over-smoothing are still maintained. Interestingly and promisingly, we find that the performance of JKNet can be promoted further if it is utilized along with our DropEdge. Actually, our DropEdge—as a flexible and general technique—is able to enhance the performance of various popular backbone networks on several benchmarks, including GCN (Kipf & Welling, 2017), ResGCN (Li et al., 2019), JKNet (Xu et al., 2018a), and GraphSAGE (Hamilton et al., 2017). We provide detailed evaluations in the experiments.

## 2 RELATED WORK

**GCNs** Inspired by the huge success of CNNs in computer vision, a large number of methods come redefining the notion of convolution on graphs under the umbrella of GCNs. The first prominent research on GCNs is presented in Bruna et al. (2013), which develops graph convolution based on spectral graph theory. Later, Kipf & Welling (2017); Defferrard et al. (2016); Henaff et al. (2015); Li et al. (2018b); Levie et al. (2017) apply improvements, extensions, and approximations on spectral-based GCNs. To address the scalability issue of spectral-based GCNs on large graphs, spatial-based GCNs have been rapidly developed (Hamilton et al., 2017; Monti et al., 2017; Niepert et al., 2016;

---

[2]To check the efficacy of DropEdge more clearly, here we have removed bias in all GCN layers, while for the experiments in § 5, the bias are kept.

Gao et al., 2018). These methods directly perform convolution in the graph domain by aggregating the information from neighbor nodes. Recently, several sampling-based methods have been proposed for fast graph representation learning, including the node-wise sampling methods (Hamilton et al., 2017), the layer-wise approach (Chen et al., 2018) and its layer-dependent variant (Huang et al., 2018). Specifically, GAT (Velickovic et al., 2018) has discussed applying dropout on edge attentions. While it actually is a post-conducted version of DropEdge before attention computation, the relation to over-smoothing is never explored in Velickovic et al. (2018). In our paper, however, we have formally presented the formulation of DropEdge and provided rigorous theoretical justification of its benefit in alleviating over-smoothing. We also carried out extensive experiments by imposing DropEdge on several popular backbones. One additional point is that we further conduct adjacency normalization after dropping edges, which, even simple, is able to make it much easier to converge during training and reduce gradient vanish as the number of layers grows.

**Deep GCNs**   Despite the fruitful progress, most previous works only focus on shallow GCNs while the deeper extension is seldom discussed. The attempt for building deep GCNs is dated back to the GCN paper (Kipf & Welling, 2017), where the residual mechanism is applied; unexpectedly, as shown in their experiments, residual GCNs still perform worse when the depth is 3 and beyond. The authors in Li et al. (2018a) first point out the main difficulty in constructing deep networks lying in over-smoothing, but unfortunately, they never propose any method to address it. The follow-up study (Klicpera et al., 2019) solves over-smoothing by using personalized PageRank that additionally involves the rooted node into the message passing loop; however, the accuracy is still observed to decrease when the depth increases from 2. JKNet (Xu et al., 2018a) employs dense connections for multi-hop message passing which is compatible with DropEdge for formulating deep GCNs. Oono & Suzuki (2019) theoretically prove that the node features of deep GCNs will converge to a subspace and incur information loss. It generalizes the conclusion in Li et al. (2018a) by further considering the ReLu function and convolution filters. Our interpretations on why DropEdge can impede over-smoothing is based on the concepts proposed by Oono & Suzuki (2019). A recent method  (Li et al., 2019) has incorporated residual layers, dense connections and dilated convolutions into GCNs to facilitate the development of deep architectures. Nevertheless, this model is targeted on graph-level classification (*i.e.* point cloud segmentation), where the data points are graphs and naturally disconnected from each other. In our task for node classification, the samples are nodes and they all couple with each other, thus the over-smoothing issue is more demanded to be addressed. By leveraging DropEdge, we are able to relieve over-smoothing, and derive more enhanced deep GCNs on node classification.

## 3   Notations and Preliminaries

**Notations.**   Let $\mathcal{G} = (\mathbb{V}, \mathcal{E})$ represent the input graph of size $N$ with nodes $v_i \in \mathbb{V}$ and edges $(v_i, v_j) \in \mathcal{E}$. The node features are denoted as $\boldsymbol{X} = \{\boldsymbol{x}_1, \cdots, \boldsymbol{x}_N\} \in \mathbb{R}^{N \times C}$, and the adjacency matrix is defined as $\boldsymbol{A} \in \mathbb{R}^{N \times N}$ which associates each edge $(v_i, v_j)$ with its element $A_{ij}$. The node degrees are given by $\boldsymbol{d} = \{d_1, \cdots, d_N\}$ where $d_i$ computes the sum of edge weights connected to node $i$. We define $\boldsymbol{D}$ as the degree matrix whose diagonal elements are obtained from $\boldsymbol{d}$.

**GCN**   is originally developed by Kipf & Welling (2017). The feed forward propagation in GCN is recursively conducted as

$$\boldsymbol{H}^{(l+1)} = \sigma\left(\hat{\boldsymbol{A}} \boldsymbol{H}^{(l)} \boldsymbol{W}^{(l)}\right), \tag{1}$$

where $\boldsymbol{H}^{(l+1)} = \{\boldsymbol{h}_1^{(l+1)}, \cdots, \boldsymbol{h}_N^{(l+1)}\}$ are the hidden vectors of the $l$-th layer with $\boldsymbol{h}_i^{(l)}$ as the hidden feature for node $i$; $\hat{\boldsymbol{A}} = \hat{\boldsymbol{D}}^{-1/2}(\boldsymbol{A} + \boldsymbol{I})\hat{\boldsymbol{D}}^{-1/2}$ is the re-normalization of the adjacency matrix, and $\hat{\boldsymbol{D}}$ is the corresponding degree matrix of $\boldsymbol{A} + \boldsymbol{I}$; $\sigma(\cdot)$ is a nonlinear function, *i.e.* the ReLu function; and $\boldsymbol{W}^{(l)} \in \mathbb{R}^{C_l \times C_{l-1}}$ is the filter matrix in the $l$-th layer with $C_l$ refers to the size of $l$-th hidden layer. We denote one-layer GCN computed by Equation 1 as Graph Convolutional Layer (**GCL**) in what follows.

## 4 OUR METHOD: DROPEDGE

This section first introduces the methodology of the DropEdge technique as well as its layer-wise variant where the adjacency matrix for each GCN layer is perturbed individually. We also explain how the proposed DropEdge can prevent over-fitting and over-smoothing in generic GCNs. Particularly for over-smoothing, we provide its mathematical definition and theoretical derivations on showing the benefits of DropEdge.

### 4.1 METHODOLOGY

At each training epoch, the DropEdge technique drops out a certain rate of edges of the input graph by random. Formally, it randomly enforces $Vp$ non-zero elements of the adjacency matrix $\mathbf{A}$ to be zeros, where $V$ is the total number of edges and $p$ is the dropping rate. If we denote the resulting adjacency matrix as $\mathbf{A}_{\mathrm{drop}}$, then its relation with $\mathbf{A}$ becomes

$$\mathbf{A}_{\mathrm{drop}} = \mathbf{A} - \mathbf{A}', \tag{2}$$

where $\mathbf{A}'$ is a sparse matrix expanded by a random subset of size $Vp$ from original edges $\mathcal{E}$. Following the idea of Kipf & Welling (2017), we also perform the re-normalization trick on $\mathbf{A}_{\mathrm{drop}}$, leading to $\hat{\mathbf{A}}_{\mathrm{drop}}$. We replace $\hat{\mathbf{A}}$ with $\hat{\mathbf{A}}_{\mathrm{drop}}$ in Equation 1 for propagation and training. When validation and testing, DropEdge is not utilized.

**Preventing over-fitting.** DropEdge produces varying perturbations of the graph connections. As a result, it generates different random deformations of the input data and can be regarded as a data augmentation skill for graphs. To explain why this is valid, we provide an intuitive understanding here. The key in GCNs is to aggregate neighbors' information for each node, which can be understood as a weighted sum of the neighbor features (the weights are associated with the edges). From the perspective of neighbor aggregation, DropEdge enables a random subset aggregation instead of the full aggregation during GNN training. Statistically, DropEdge only changes the expectation of the neighbor aggregation up to a multiplier $p$, if we drop edges with probability $p$. This multiplier will be actually removed after weights normalization, which is often the case in practice. Therefore, DropEdge does not change the expectation of neighbor aggregation and is an unbiased data augmentation technique for GNN training, similar to typical image augmentation skills (*e.g.* rotation, cropping and flapping) that are capable of hindering over-fitting in training CNNs. We will provide experimental validations in § 5.1.

**Layer-Wise DropEdge.** The above formulation of DropEdge is one-shot with all layers sharing the same perturbed adjacency matrix. Indeed, we can perform DropEdge for each individual layer. Specifically, we obtain $\hat{\mathbf{A}}_{\mathrm{drop}}^{(l)}$ by independently computing Equation 2 for each $l$-th layer. Different layer could have different adjacency matrix $\hat{\mathbf{A}}_{\mathrm{drop}}^{(l)}$. Such layer-wise version brings in more randomness and deformations of the original data, and we will experimentally compare its performance with the original DropEdge in § 5.2.

Over-smoothing is another obstacle of training deep GCNs, and we will detail how DropEdge can address it to some extent in the next section. For simplicity, the following derivations assume all GCLs share the same perturbed adjacency matrix, and we will leave the discussion on layer-wise DropEdge for future exploration.

### 4.2 TOWARDS PREVENTING OVER-SMOOTHING

By its original definition in Li et al. (2018a), the over-smoothing phenomenon implies that the node features will converge to a fixed point as the network depth increases. This unwanted convergence restricts the output of deep GCNs to be only relevant to the graph topology but independent to the input node features, which as a matter of course incurs detriment of the expressive power of GCNs. Oono & Suzuki (2019) has generalized the idea in Li et al. (2018a) by taking both the non-linearity (*i.e.* the ReLu function) and the convolution filters into account; they explain over-smoothing as convergence to a subspace rather than convergence to a fixed point. This paper will use the concept of subspace by Oono & Suzuki (2019) for more generality.

We first provide several relevant definitions that facilitate our later presentations.

**Definition 1** (subspace). *Let $\mathcal{M} := \{\boldsymbol{E}\boldsymbol{C} | \boldsymbol{C} \in \mathbb{R}^{M \times C}\}$ be an $M$-dimensional subspace in $\mathbb{R}^{N \times C}$, where $\boldsymbol{E} \in \mathbb{R}^{N \times M}$ is orthogonal, i.e. $\boldsymbol{E}^{\mathrm{T}}\boldsymbol{E} = \boldsymbol{I}_M$, and $M \leq N$.*

**Definition 2** ($\epsilon$-smoothing). *We call the $\epsilon$-smoothing of node features happens for a GCN, if all its hidden vectors $\boldsymbol{H}^{(l)}$ beyond a certain layer $L$ have a distance no larger than $\epsilon$ ($\epsilon > 0$) with respect to a subspace $\mathcal{M}$ that is independent to the input features, namely,*

$$d_{\mathcal{M}}(\boldsymbol{H}^{(l)}) < \epsilon, \forall l \geq L, \tag{3}$$

*where $d_{\mathcal{M}}(\cdot)$ computes the distance between the input matrix and the subspace $\mathcal{M}$.*[3]

**Definition 3** (the $\epsilon$-smoothing layer). *Given the subspace $\mathcal{M}$ and $\epsilon$, we call the minimal value of the layers that satisfy Equation 3 as the $\epsilon$-smoothing layer, that is, $l^*(\mathcal{M}, \epsilon) := \min_l\{d_{\mathcal{M}}(\boldsymbol{H}^{(l)}) < \epsilon\}$.*

Since conducting analysis exactly based on the $\epsilon$-smoothing layer is difficult, we instead define the relaxed $\epsilon$-smoothing layer which is proved to be an upper bound of $l^*$.

**Definition 4** (the relaxed $\epsilon$-smoothing layer). *Given the subspace $\mathcal{M}$ and $\epsilon$, we call $\hat{l}(\mathcal{M}, \epsilon) = \lceil \frac{\log(\epsilon/d_{\mathcal{M}}(\boldsymbol{X}))}{\log s\lambda} \rceil$ as the relaxed smoothing layer, where, $\lceil \cdot \rceil$ computes the ceil of the input, $s$ is the supremum of the filters' singular values over all layers, and $\lambda$ is the second largest eigenvalue of $\hat{\boldsymbol{A}}$. Besides, we have $\hat{l} \geq l^*$.*[4]

According to the conclusions by the authors in Oono & Suzuki (2019), a sufficiently deep GCN will certainly suffer from the $\epsilon$-smoothing issue for any small value of $\epsilon$ under some mild conditions (the details are included in the supplementary material). Note that they only prove the existence of $\epsilon$-smoothing in deep GCN without developing any method to address it.

Here, we will demonstrate that adopting DropEdge alleviates the $\epsilon$-smoothing issue in two aspects: **1.** By reducing node connections, DropEdge is proved to slow down the convergence of over-smoothing; in other words, the value of the relaxed $\epsilon$-smoothing layer will only increase if using DropEdge. **2.** The gap between the dimensions of the original space and the converging subspace, *i.e.* $N - M$ measures the amount of information loss; larger gap means more severe information loss. As shown by our derivations, DropEdge is able to increase the dimension of the converging subspace, thus capable of reducing information loss.

We summarize our conclusions as follows.

**Theorem 1.** *We denote the original graph as $\mathcal{G}$ and the one after dropping certain edges out as $\mathcal{G}'$. Given a small value of $\epsilon$, we assume $\mathcal{G}$ and $\mathcal{G}'$ will encounter the $\epsilon$-smoothing issue with regard to subspaces $\mathcal{M}$ and $\mathcal{M}'$, respectively. Then, either of the following inequalities holds after sufficient edges removed.*

- *The relaxed smoothing layer only increases: $\hat{l}(\mathcal{M}, \epsilon) \leq \hat{l}(\mathcal{M}', \epsilon)$;*
- *The information loss is decreased: $N - dim(\mathcal{M}) > N - dim(\mathcal{M}')$.*

The proof of Theorem 1 is based on the derivations in Oono & Suzuki (2019) as well as the concept of *mixing time* that has been studied in the random walk theory (Lovász et al., 1993). We provide the full details in the supplementary material. Theorem 1 tells that DropEdge either reduces the convergence speed of over-smoothing or relieves the information loss caused by it. In this way, DropEdge enables us to train deep GCNs more effectively.

### 4.3 DISCUSSIONS

This sections contrasts the difference between DropEdge and other related concepts including Dropout, DropNode, and Graph Sparsification.

**DropEdge vs. Dropout** The Dropout trick (Hinton et al., 2012) is trying to perturb the feature matrix by randomly setting feature dimensions to be zeros, which may reduce the effect of over-fitting but is of no help to preventing over-smoothing since it does not make any change of the adjacency

---

[3]The definition of $d_{\mathcal{M}}(\cdot)$ is provided in the supplementary material.

[4] All detailed definitions and proofs are provided in the appendix.

matrix. As a reference, DropEdge can be regarded as a generation of Dropout from dropping feature dimensions to dropping edges, which mitigates both over-fitting and over-smoothing. In fact, the impacts of Dropout and DropEdge are complementary to each other, and their compatibility will be shown in the experiments.

**DropEdge vs. DropNode** Another related vein belongs to the kind of node sampling based methods, including GraphSAGE (Hamilton et al., 2017), FastGCN (Chen et al., 2018), and AS-GCN (Huang et al., 2018). We name this category of approaches as DropNode. For its original motivation, DropNode samples sub-graphs for mini-batch training, and it can also be treated as a specific form of dropping edges since the edges connected to the dropping nodes are also removed. However, the effect of DropNode on dropping edges is node-oriented and indirect. By contrast, DropEdge is edge-oriented, and it is possible to preserve all node features for the training (if they can be fitted into the memory at once), exhibiting more flexibility. Further, to maintain desired performance, the sampling strategies in current DropNode methods are usually inefficient, for example, GraphSAGE suffering from the exponentially-growing layer size, and AS-GCN requiring the sampling to be conducted recursively layer by layer. Our DropEdge, however, neither increases the layer size as the depth grows nor demands the recursive progress because the sampling of all edges are parallel.

**DropEdge vs. Graph-Sparsification** Graph-Sparsification (Eppstein et al., 1997) is an old research topic in the graph domain. Its optimization goal is removing unnecessary edges for graph compressing while keeping almost all information of the input graph. This is clearly district to the purpose of DropEdge where no optimization objective is needed. Specifically, DropEdge will remove the edges of the input graph by random at each training time, whereas Graph-Sparsification resorts to a tedious optimization method to determine which edges to be deleted, and once those edges are discarded the output graph keeps unchanged.

## 5 EXPERIMENTS

**Datasets** Joining the previous works' practice, we focus on four benchmark datasets varying in graph size and feature type: (1) classifying the research topic of papers in three citation datasets: Cora, Citeseer and Pubmed (Sen et al., 2008); (2) predicting which community different posts belong to in the Reddit social network (Hamilton et al., 2017). Note that the tasks in Cora, Citeseer and Pubmed are transductive underlying all node features are accessible during training, while the task in Reddit is inductive meaning the testing nodes are unseen for training. We apply the full-supervised training fashion used in Huang et al. (2018) and Chen et al. (2018) on all datasets in our experiments. The statics of all datasets are listed in the supplemental materials.

### 5.1 CAN DROPEDGE GENERALLY IMPROVE THE PERFORMANCE OF DEEP GCNS?

In this section, we are interested in if applying DropEdge can promote the performance of current popular GCNs (especially their deep architectures) on node classification.

**Implementations** We consider five backbones: GCN (Kipf & Welling, 2017), ResGCN (He et al., 2016; Li et al., 2019), JKNet (Xu et al., 2018a), IncepGCN[5] and GraphSAGE (Hamilton et al., 2017) with varying depth from 2 to 64.[6] Since different structure exhibits different training dynamics on different dataset, to enable more robust comparisons, we perform random hyper-parameter search for each model, and report the case giving the best accuracy on validation set of each benchmark. The searching space of hyper-parameters and more details are provided in Table 4 in the supplementary material. Regarding the same architecture w or w/o DropEdge, we apply the same set of hyper-parameters except the drop rate $p$ for fair evaluation.

**Overall Results** Table 1 summaries the results on all datasets. We only report the performance of the model with 2/8/32 layers here due to the space limit, and provide the accuracy under other different depths in the supplementary material. It's observed that DropEdge consistently improves the

---

[5]The formulation is given in the appendix.
[6]For Reddit, the maximum depth is 32 considering the memory bottleneck.

Table 1: Testing accuracy (%) comparisons on different backbones w and w/o DropEdge.

| Dataset | Backbone | 2 layers | | 8 layers | | 32 layers | |
|---|---|---|---|---|---|---|---|
| | | Orignal | DropEdge | Orignal | DropEdge | Orignal | DropEdge |
| Cora | GCN | 86.10 | **86.50** | 78.70 | **85.80** | 71.60 | **74.60** |
| | ResGCN | - | - | 85.40 | **86.90** | 85.10 | **86.80** |
| | JKNet | - | - | 86.70 | **87.80** | 87.10 | **87.60** |
| | IncepGCN | - | - | 86.70 | **88.20** | 87.40 | **87.70** |
| | GraphSAGE | 87.80 | **88.10** | 84.30 | **87.10** | 31.90 | **32.20** |
| Citeseer | GCN | 75.90 | **78.70** | 74.60 | **77.20** | 59.20 | **61.40** |
| | ResGCN | - | - | 77.80 | **78.80** | 74.40 | **77.90** |
| | JKNet | - | - | 79.20 | **80.20** | 71.70 | **80.00** |
| | IncepGCN | - | - | 79.60 | **80.50** | 72.60 | **80.30** |
| | GraphSAGE | 78.40 | **80.00** | 74.10 | **77.10** | 37.00 | **53.60** |
| Pubmed | GCN | 90.20 | **91.20** | 90.10 | **90.90** | 84.60 | **86.20** |
| | ResGCN | - | - | 89.60 | **90.50** | 90.20 | **91.10** |
| | JKNet | - | - | 90.60 | **91.20** | 89.20 | **91.30** |
| | IncepGCN | - | - | 90.20 | **91.50** | OOM | **90.50** |
| | GraphSAGE | 90.10 | **90.70** | 90.20 | **91.70** | 41.30 | **47.90** |
| Reddit | GCN | 96.11 | **96.13** | 96.17 | **96.48** | 45.55 | **50.51** |
| | ResGCN | - | - | 96.37 | **96.46** | 93.93 | **94.27** |
| | JKNet | - | - | 96.82 | **97.02** | OOM | OOM |
| | IncepGCN | - | - | 96.43 | **96.87** | OOM | OOM |
| | GraphSAGE | 96.22 | **96.28** | 96.38 | **96.42** | 96.43 | **96.47** |

testing accuracy for all cases. The improvement is more clearly depicted in Figure 2a, where we have computed the average absolute improvement over all backbones by DropEdge on each dataset under different numbers of layers. On Citeseer, for example, DropEdge yields further improvement for deeper architecture; it gains 0.9% average improvement for the model with 2 layers while achieving a remarkable 13.5% increase for the model with 64 layers. In addition, the validation losses of all 4-layer models on Cora are shown in Figure 2b. The curves along the training epoch are dramatically pulled down after applying DropEdge, which also explains the effect of DropEdge on alleviating over-fitting. Another valuable observation in Table 1 is that the 32-layer IncepGCN without DropEdge incurs the Out-Of-Memory (OOM) issue while the model with DropEdge survives, showing the advantage of DropEdge to save memory consuming by making the adjacency matrix sparse.

**Comparison with SOTAs**  We select the best performance for each backbone with DropEdge, and contrast them with existing State of the Arts (SOTA), including GCN, FastGCN, AS-GCN and GraphSAGE in Table 2; for the SOTA methods, we reuse the results reported in Huang et al. (2018). We have these findings: (1) Clearly, our DropEdge obtains significant enhancement against SOTAs; particularly on Reddit, the best accuracy by our method is 97.02%, and it is better than the previous best by AS-GCN (96.27%), which is regarded as a remarkable boost considering the challenge on this benchmark. (2) For most models with DropEdge, the best accuracy is obtained under the depth beyond 2, which again verifies the impact of DropEdge on formulating deep networks. (3) As mentioned in § 4.3, FastGCN, AS-GCN and GraphSAGE are considered as the DropNode extensions of GCNs. The DropEdge based approaches outperform the DropNode based variants as shown in Table 2, which somehow confirms the effectiveness of DropEdge. Actually, employing DropEdge upon the DropNode methods further delivers promising enhancement, which can be checked by revisiting the increase by DropEdge for GraphSAGE in Table 1.

## 5.2 How does DropEdge help?

This section continues a more in-depth analysis on DropEdge and attempts to figure out why it works. Due to the space limit, we only provide the results on Cora, and defer the evaluations on other datasets to the supplementary material.

Note that this section mainly focuses on analyzing DropEdge and its variants, without the concern with pushing state-of-the-art results. So, we do not perform delicate hyper-parameter selection. We employ GCN as the backbone in this section. Here, GCN-$n$ denotes GCN of depth $n$. The hidden dimension, learning rate and weight decay are fixed to 256, 0.005 and 0.0005, receptively. The

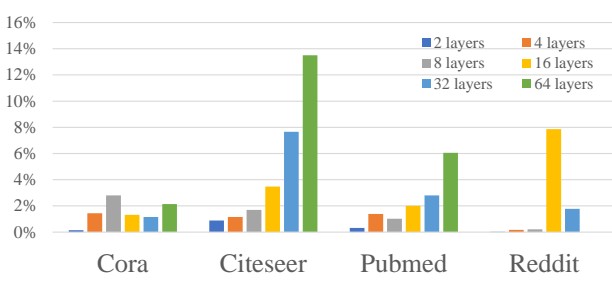 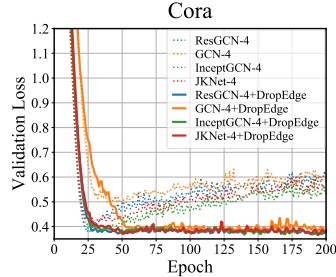

(a) The average absolute improvement by DropEdge.

(b) The validation loss on different backbones w and w/o DropEdge.

Figure 2

Table 2: Accuracy (%) comparisons with SOTAs. The number in parenthesis denotes the network depth for the models with DropEdge.

|  | Transductive | | | Inductive |
| --- | --- | --- | --- | --- |
|  | Cora | Citeseer | Pubmed | Reddit |
| GCN | 86.64 | 79.34 | 90.22 | 95.68 |
| FastGCN | 85.00 | 77.60 | 88.00 | 93.70 |
| ASGCN | 87.44 | 79.66 | 90.60 | 96.27 |
| GraphSAGE | 82.20 | 71.40 | 87.10 | 94.32 |
| GCN+DropEdge | 87.60(4) | 79.20(4) | 91.30(4) | 96.71(4) |
| ResGCN+DropEdge | 87.00(4) | 79.40(16) | 91.10(32) | 96.48(16) |
| JKNet+DropEdge | 88.00(16) | 80.20(8) | 91.60(64) | **97.02(8)** |
| IncepGCN+DropEdge | **88.20(8)** | **80.50(8)** | 91.60(4) | 96.87(8) |
| GraphSAGE+DropEdge | 88.10(4) | 80.00(2) | **91.70(8)** | 96.54(4) |

random seed is fixed. We train all models with 200 epochs. Unless otherwise mentioned, we do not utilize the "withloop" and "withbn" operation (see their definitions in Table 4 in the appendix).

### 5.2.1 ON PREVENTING OVER-SMOOTHING

As discussed in § 4.2, the over-smoothing issue exists when the top-layer outputs of GCN converge to a subspace and become unrelated to the input features with the increase in depth. Since we are unable to derive the converging subspace explicitly, we measure the degree of over-smoothing by instead computing the difference between the output of the current layer and that of the previous one. We adopt the Euclidean distance for the difference computation. Lower distance means more serious over-smoothing. Experiments are conducted on GCN-8.

Figure 3 (a) shows the distances of different intermediate layers (from 2 to 6) under different edge dropping rates (0 and 0.8). Clearly, over-smoothing becomes more serious in GCN as the layer grows, which is consistent with our conjecture. Conversely, the model with DropEdge ($p = 0.8$) reveals higher distance and slower convergent speed than that without DropEdge ($p = 0$), implying the importance of DropEdge to alleviating over-smoothing. We are also interested in how the over-smoothing will act after training. For this purpose, we display the results after 150-epoch training in Figure 3 (b). For GCN without DropEdge, the difference between outputs of the 5-th and 6-th layers is equal to 0, indicating that the hidden features have converged to a certain stationary point. On the contrary, GCN with DropEdge performs promisingly, as the distance does not vanish to zero when

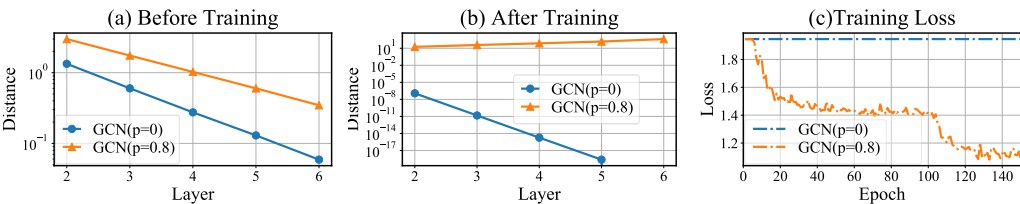

Figure 3: Analysis on over-smoothing. Smaller distance means more serious over-smoothing.

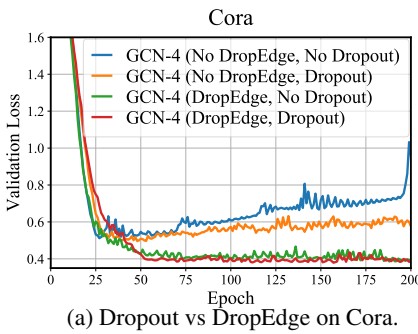
(a) Dropout vs DropEdge on Cora.

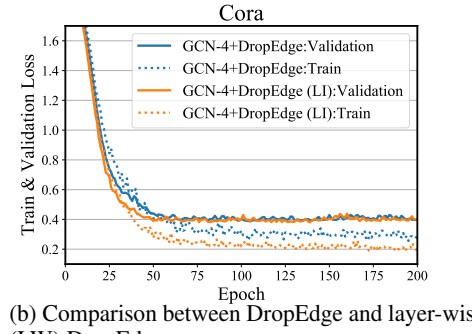
(b) Comparison between DropEdge and layer-wise (LW) DropEdge.

Figure 4

the number of layers grows; it probably has successfully learned meaningful node representations after training, which could also be validated by the training loss in Figure 3 (c).

### 5.2.2    ON COMPATIBILITY WITH DROPOUT

§ 4.3 has discussed the difference between DropEdge and Dropout. Hence, we conduct an ablation study on GCN-4, and the validation losses are demonstrated in Figure 4a. It reads that while both Dropout and DropEdge are able to facilitate the training of GCN, the improvement by DropEdge is more significant, and if we adopt them concurrently, the loss is decreased further, indicating the compatibility of DropEdge with Dropout.

### 5.2.3    ON LAYER-WISE DROPEDGE

§ 4.1 has descried the Layer-Wise (LW) extension of DropEdge. Here, we provide the experimental evaluation on assessing its effect. As observed from Figure 4b, the LW DropEdge achieves lower training loss than the original version, whereas the validation value between two models is comparable. It implies that LW DropEdge can facilitate the training further than original DropEdge. However, we prefer to use DropEdge other than the LW variant so as to not only avoid the risk of over-fitting but also reduces computational complexity since LW DropEdge demands to sample each layer and spends more time.

## 6    CONCLUSION

We have presented DropEdge, a novel and efficient technique to facilitate the development of deep Graph Convolutional Networks (GCNs). By dropping out a certain rate of edges by random, DropEdge includes more diversity into the input data to prevent over-fitting, and reduces message passing in graph convolution to alleviate over-smoothing. Considerable experiments on Cora, Citeseer, Pubmed and Reddit have verified that DropEdge can generally and consistently promote the performance of current popular GCNs, such as GCN, ResGCN, JKNet, IncepGCN, and GraphSAGE. It is expected that our research will open up a new venue on a more in-depth exploration of deep GCNs for broader potential applications.

## 7    ACKNOWLEDGEMENTS

This research was funded by National Science and Technology Major Project of the Ministry of Science and Technology of China (No. 2018AAA0102900). Finally, Yu Rong wants to thank, in particular, the invaluable love and support from Yunman Huang over the years. Will you marry me?

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

## A   APPENDIX: PROOF OF THEOREM 1

To prove Theorem 1, we need to borrow the following definitions and corollaries from Oono & Suzuki (2019). First, we denote the maximum singular value of $\boldsymbol{W}_l$ by $s_l$ and set $s := \sup_{l \in \mathbb{N}_+} s_l$. We assume that $\boldsymbol{W}_l$ of all layers are initialized so that $s \leq 1$. Second, we denote the distance that induced as the Frobenius norm from $\boldsymbol{X}$ to $\mathcal{M}$ by $d_{\mathcal{M}}(\boldsymbol{X}) := \inf_{\boldsymbol{Y} \in \mathcal{M}} ||\boldsymbol{X} - \boldsymbol{Y}||_{\mathrm{F}}$. Then, we recall Corollary 3 and Proposition 1 in Oono & Suzuki (2019) as Corollary 1 below.

**Corollary 1.** *Let $\lambda_1 \leq \cdots \leq \lambda_N$ be the eigenvalues of $\hat{\boldsymbol{A}}$, sorted in ascending order. Suppose the multiplicity of the largest eigenvalue $\lambda_N$ is $M(\leq N)$, i.e., $\lambda_{N-M} < \lambda_{N-M+1} = \cdots = \lambda_N$ and the second largest eigenvalue is defined as*

$$\lambda := \max_{n=1}^{N-M} |\lambda_n| < |\lambda_N|. \tag{4}$$

*Let $\boldsymbol{E}$ to be the eigenspace associated with $\lambda_{N-M+1}, \cdots, \lambda_N$. Then we have $\lambda < \lambda_N = 1$, and*

$$d_{\mathcal{M}}(\boldsymbol{H}^{(l)}) \leq s_l \lambda d_{\mathcal{M}}(\boldsymbol{H}^{(l-1)}), \tag{5}$$

*where $\mathcal{M} := \{\boldsymbol{EC}|\boldsymbol{C} \in \mathbb{R}^{M \times C}\}$. Besides, $s_l \lambda < 1$, implying that the output of the l-th layer of GCN on $\mathcal{G}$ exponentially approaches $\mathcal{M}$.*

We also need to adopt some concepts from Lovász et al. (1993) in proving Theorem 1. Consider the graph $\mathcal{G}$ as an electrical network, where each edge represents an unit resistance. Then the effective resistance, $R_{st}$ from node $s$ to node $t$ is defined as the total resistance between node $s$ and $t$. According to Corollary 3.3 and Theorem 4.1 (i) in Lovász et al. (1993), we can build the connection between $\lambda$ and $R_{st}$ for each connected component via commute time as the following inequality.

$$\lambda \geq 1 - \frac{1}{R_{st}}(\frac{1}{d_s} + \frac{1}{d_t}). \tag{6}$$

Prior to proving Theorem 1, we first derive the lemma below.

**Lemma 2.** *The $\epsilon$-smoothing happens whenever the layer number satisfies*

$$l \geq \hat{l} = \lceil \frac{\log \frac{\epsilon}{d_{\mathcal{M}}(\boldsymbol{X})}}{\log(s\lambda)} \rceil, \tag{7}$$

*where $\lceil \cdot \rceil$ computes the ceil of the input. It means $\hat{l} \geq l^*$.*

*Proof.* We start our proof from Inequality 5, leading to

$$d_{\mathcal{M}}(\boldsymbol{H}^{(l)}) \leq s_l \lambda d_{\mathcal{M}}(\boldsymbol{H}^{(l-1)})$$
$$\leq (\prod_{i=1}^{l} s_i)\lambda^l d_{\mathcal{M}}(\boldsymbol{X})$$
$$\leq s^l \lambda^l d_{\mathcal{M}}(\boldsymbol{X})$$

When it reaches $\epsilon$-smoothing, the following inequality should be satisfied as

$$d_{\mathcal{M}}(\boldsymbol{H}^{(l)}) \leq s^l \lambda^l d_{\mathcal{M}}(\boldsymbol{X}) < \epsilon,$$
$$\Rightarrow l \log s\lambda < \log \frac{\epsilon}{d_{\mathcal{M}}(\boldsymbol{X})}. \tag{8}$$

Since $0 \leq s\lambda < 1$, then $\log s\lambda < 0$. Therefore, the Inequality 8 becomes

$$l > \frac{\log \frac{\epsilon}{d_{\mathcal{M}}(\boldsymbol{X})}}{\log s\lambda}. \tag{9}$$

Clearly, we have $\hat{l} \geq l^*$ since $l^*$ is defined as the minimal layer that satisfies $\epsilon$-smoothing. The proof is concluded. □

Now, we prove Theorem 1.

*Proof.* Our proof relies basically on the connection between $\lambda$ and $R_{st}$ in Equation (6). We recall Corollary 4.3 in Lovász et al. (1993) that removing any edge from $\mathcal{G}$ can only increase any $R_{st}$, then according to (6), the lower bound of $\lambda$ only increases if the removing edge is not connected to either $s$ or $t$ (*i.e.* the degree $d_s$ and $d_t$ keep unchanged). Since there must exist a node pair satisfying $R_{st} = \infty$ after sufficient edges (except self-loops) are removed from one connected component of $\mathcal{G}$, we have the infinite case $\lambda = 1$ given in Equation (6) that both $1/d_s$ and $1/d_t$ are consistently bounded by a finite number,*i.e.* 1. It implies $\lambda$ does increase before it reaches $\lambda = 1$. As $\hat{l}$ is positively related to $\lambda$ (see the right side of Equation (7) where $log(s\lambda)$ <0), we have proved the first part of Theorem 1, *i.e.*, $\hat{l}(\mathcal{M}, \epsilon) \leq \hat{l}(\mathcal{M}', \epsilon)$ after removing sufficient edges.

When there happens $R_{st} = \infty$, the connected component is disconnected into two parts, which leads to the increment of the dimension of $\mathcal{M}$ by 1 and proves the second part of Theorem 1. i.e., *the information loss is decreased:* $N - dim(\mathcal{M}) > N - dim(\mathcal{M}')$.

$\square$

# B  APPENDIX: MORE DETAILS IN EXPERIMENTS

## B.1  DATASETS STATISTICS

**Datasets**  The statistics of all datasets are summarized in Table 3.

Table 3: Dataset Statistics

| Datasets | Nodes | Edges | Classes | Features | Traing/Validation/Testing | Type |
|---|---|---|---|---|---|---|
| Cora | 2,708 | 5,429 | 7 | 1,433 | 1,208/500/1,000 | Transductive |
| Citeseer | 3,327 | 4,732 | 6 | 3,703 | 1,812/500/1,000 | Transductive |
| Pubmed | 19,717 | 44,338 | 3 | 500 | 18,217/500/1,000 | Transductive |
| Reddit | 232,965 | 11,606,919 | 41 | 602 | 152,410/23,699/55,334 | Inductive |

## B.2  MODELS AND BACKBONES

**Backbones**  Other than the multi-layer GCN, we replace the CNN layer with graph convolution layer to implement three popular backbones recast from image classification. They are residual network (ResGCN)(He et al., 2016; Li et al., 2019), inception network (IncepGCN)(Szegedy et al., 2016) and dense network (JKNet) (Huang et al., 2017; Xu et al., 2018b). Figure 5 shows the detailed architectures of four backbones. Furthermore, we employ one input GCL and one output GCL on these four backbones. Therefore, the layers in ResGCN, JKNet and InceptGCN are at least 3 layers. All backbones are implemented in Pytorch (Paszke et al., 2017). For GraphSAGE, we utilize the Pytorch version implemented by DGL(Wang et al., 2019).

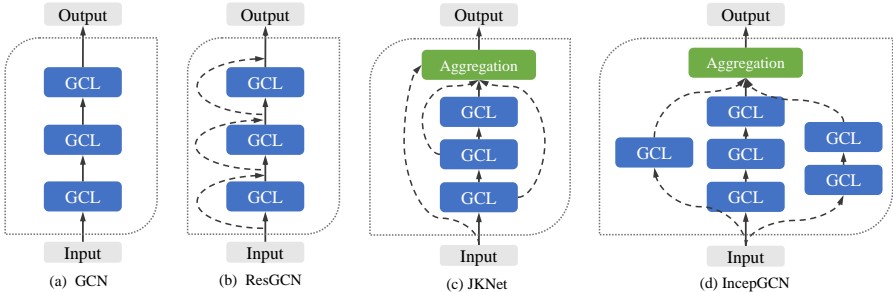

Figure 5: The illustration of four backbones. GCL indicates graph convolutional layer.

**Self Feature Modeling**  We also implement a variant of graph convolution layer with self feature modeling (Fout et al., 2017):

$$\mathbf{H}^{(l+1)} = \sigma\left(\hat{\mathbf{A}}\mathbf{H}^{(l)}\mathbf{W}^{(l)} + \mathbf{H}^{(l)}\mathbf{W}^{(l)}_{\text{self}}\right), \tag{10}$$

where $\mathbf{W}^{(l)}_{\text{self}} \in \mathbb{R}^{C_l \times C_{l-1}}$.

**Hyper-parameter Optimization**  We adopt the Adam optimizer for model training. To ensure the re-productivity of the results, the seeds of the random numbers of all experiments are set to the same. We fix the number of training epoch to $400$ for all datasets. All experiments are conducted on a NVIDIA Tesla P40 GPU with 24GB memory.

Given a model with $n \in \{2, 4, 8, 16, 32, 64\}$ layers, the hidden dimension is $128$ and we conduct a random search strategy to optimize the other hyper-parameter for each backbone in § 5.1. The decryptions of hyper-parameters are summarized in Table 4. Table 5 depicts the types of the normalized adjacency matrix that are selectable in the "normalization" hyper-parameter. For GraphSAGE, the aggregation type like GCN, MAX, MEAN, or LSTM is a hyper-parameter as well.

For each model, we try 200 different hyper-parameter combinations via random search and select the best test accuracy as the result. Table 6 summaries the hyper-parameters of each backbone with the best accuracy on different datasets and their best accuracy are reported in Table 2.

## B.3 THE VALIDATION LOSS ON DIFFERENT BACKBONES W AND W/O DROPEDGE.

Figure 6 depicts the additional results of validation loss on different backbones w and w/o DropEdge.

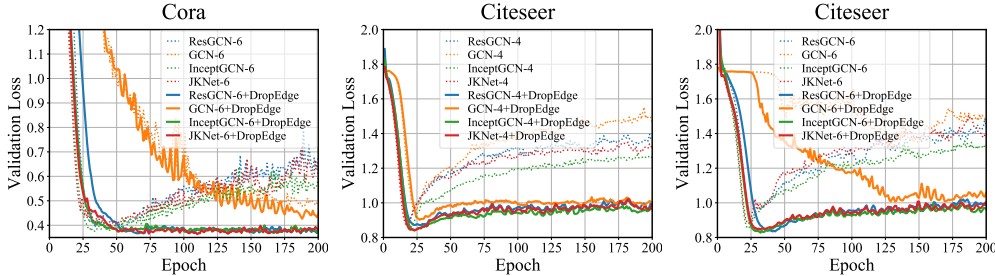

Figure 6: The validation loss on different backbones w and w/o DropEdge. GCN-$n$ denotes PlainGCN of depth $n$; similar denotation follows for other backbones.

## B.4 THE ABLATION STUDY ON CITESEER

Figure 7a shows the ablation study of Dropout vs. DropEdge and Figure 4b depicts a comparison between the proposed DropEdge and the layer-wise DropEdge on Citeseer.

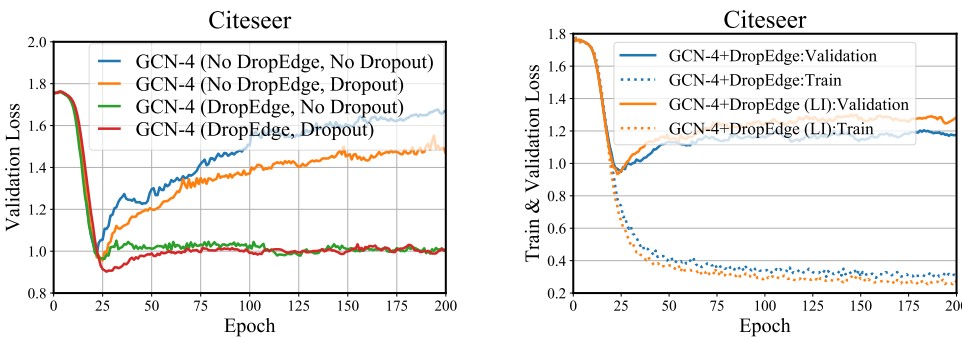

(a) Ablation study of Dropout vs. DropEdge on Citeseer.

(b) Performance comparison of layer-wise DropEdge.

Figure 7

Table 4: Hyper-parameter Description

| Hyper-parameter | Description |
| --- | --- |
| lr | learning rate |
| weight-decay | L2 regulation weight |
| sampling-percent | edge preserving percent $(1 - p)$ |
| dropout | dropout rate |
| normalization | the propagation models (Kipf & Welling, 2017) |
| withloop | using self feature modeling |
| withbn | using batch normalization |

Table 5: The normalization / propagation models

| Description | Notation | $A'$ |
|---|---|---|
| First-order GCN | FirstOrderGCN | $I + D^{-1/2}AD^{-1/2}$ |
| Augmented Normalized Adjacency | AugNormAdj | $(D+I)^{-1/2}(A+I)(D+I)^{-1/2}$ |
| Augmented Normalized Adjacency with Self-loop | BingGeNormAdj | $I + (D+I)^{-1/2}(A+I)(D+I)^{-1/2}$ |
| Augmented Random Walk | AugRWalk | $(D+I)^{-1}(A+I)$ |

Table 6: The hyper-parameters of best accuracy for each backbone on all datasets.

| Dataset | Backbone | nlayers | Acc. | Hyper-parameters |
|---|---|---|---|---|
| Cora | GCN | 4 | 0.876 | lr:0.010, weight-decay:5e-3, sampling-percent:0.7, dropout:0.8, normalization:FirstOrderGCN |
| | ResGCN | 4 | 0.87 | lr:0.001, weight-decay:1e-5, sampling-percent:0.1, dropout:0.5, normalization:FirstOrderGCN |
| | JKNet | 16 | 0.88 | lr:0.008, weight-decay:5e-4, sampling-percent:0.2, dropout:0.8, normalization:AugNormAdj |
| | IncepGCN | 8 | 0.882 | lr:0.010, weight-decay:1e-3, sampling-percent:0.05, dropout:0.5, normalization:AugNormAdj |
| | GraphSage | 4 | 0.881 | lr:0.010, weight-decay:5e-4, sampling-percent:0.4, dropout:0.5, aggregator:mean |
| Citeseer | GCN | 4 | 0.792 | lr:0.009, weight-decay:1e-3, sampling-percent:0.05, dropout:0.8, normalization:BingGeNormAdj, withloop, withbn |
| | ResGCN | 16 | 0.794 | lr:0.001, weight-decay:5e-3, sampling-percent:0.5, dropout:0.3, normalization:BingGeNormAdj, withloop |
| | JKNet | 8 | 0.802 | lr:0.004, weight-decay:5e-5, sampling-percent:0.6, dropout:0.3, normalization:AugNormAdj, withloop |
| | IncepGCN | 8 | 0.805 | lr:0.002, weight-decay:5e-3, sampling-percent:0.2, dropout:0.5, normalization:BingGeNormAdj, withloop |
| | GraphSage | 2 | 0.8 | lr:0.001, weight-decay:1e-4, sampling-percent:0.1, dropout:0.5, aggregator:mean |
| Pubmed | GCN | 4 | 0.913 | lr:0.010, weight-decay:1e-3, sampling-percent:0.3, dropout:0.5, normalization:BingGeNormAdj, withloop, withbn |
| | ResGCN | 32 | 0.911 | lr:0.003, weight-decay:5e-5, sampling-percent:0.7, dropout:0.8, normalization:AugNormAdj, withloop, withbn |
| | JKNet | 64 | 0.916 | lr:0.005, weight-decay:1e-4, sampling-percent:0.5, dropout:0.8, normalization:AugNormAdj, withloop,withbn |
| | IncepGCN | 4 | 0.916 | lr:0.002, weight-decay:1e-5, sampling-percent:0.5, dropout:0.8, normalization:BingGeNormAdj, withloop, withbn |
| | GraphSage | 8 | 0.917 | lr:0.007, weight-decay:1e-4, sampling-percent:0.8, dropout:0.3, aggregator:mean |
| Reddit | GCN | 4 | 0.9671 | lr:0.005, weight-decay:1e-4, sampling-percent:0.6, dropout:0.5, normalization:AugRWalk, withloop |
| | ResGCN | 16 | 0.9648 | lr:0.009, weight-decay:1e-5, sampling-percent:0.2, dropout:0.5, normalization:BingGeNormAdj, withbn |
| | JKNet | 8 | 0.9702 | lr:0.010, weight-decay:5e-5, sampling-percent:0.6, dropout:0.5, normalization:BingGeNormAdj, withloop,withbn |
| | IncepGCN | 8 | 0.9687 | lr:0.008, weight-decay:1e-4, sampling-percent:0.4, dropout:0.5, normalization:FirstOrderGCN, withbn |
| | GraphSAGE | 4 | 0.9654 | lr:0.005, weight-decay:5e-5, sampling-percent:0.2, dropout:0.3, aggregator:mean |

Table 7: Accuracy (%) comparisons on different backbones with and without DropEdge

| Dataset | Backbone | 2 | | 4 | | 8 | | 16 | | 32 | | 64 | |
|---|---|---|---|---|---|---|---|---|---|---|---|---|---|
| | | Orignal | DropEdge | Orignal | DropEdge | Orignal | DropEdge | Orignal | DropEdge | Orignal | DropEdge | Orignal | DropEdge |
| Cora | GCN | 86.10 | 86.50 | 85.50 | 87.60 | 78.70 | 85.80 | 82.10 | 84.30 | 71.60 | 74.60 | 52.00 | 53.20 |
| | ResGCN | - | - | 86.00 | 87.00 | 85.40 | 86.90 | 85.30 | 86.90 | 85.10 | 86.80 | 79.80 | 84.80 |
| | JKNet | - | - | 86.90 | 87.70 | 86.70 | 87.80 | 86.20 | 88.00 | 87.10 | 87.60 | 86.30 | 87.90 |
| | IncepGCN | 87.80 | 88.10 | 85.60 | 87.90 | 86.70 | 88.20 | 87.10 | 87.70 | 87.40 | 87.70 | 85.30 | 88.20 |
| | GraphSAGE | - | - | 87.10 | 88.10 | 84.30 | 87.10 | 84.10 | 84.50 | 31.90 | 32.20 | 31.90 | 31.90 |
| Citeseer | GCN | 75.90 | 78.70 | 76.70 | 79.20 | 74.60 | 77.20 | 65.20 | 76.80 | 59.20 | 61.40 | 44.60 | 45.60 |
| | ResGCN | - | - | 78.90 | 78.80 | 77.80 | 78.80 | 78.20 | 79.40 | 74.40 | 77.90 | 21.20 | 75.30 |
| | JKNet | - | - | 79.10 | 80.20 | 79.20 | 80.20 | 78.80 | 80.10 | 71.70 | 80.00 | 76.70 | 80.00 |
| | IncepGCN | 78.40 | 80.00 | 79.50 | 79.90 | 79.60 | 80.50 | 78.50 | 80.20 | 72.60 | 80.30 | 79.00 | 79.90 |
| | GraphSAGE | - | - | 77.30 | 79.20 | 74.10 | 77.10 | 72.90 | 74.50 | 37.00 | 53.60 | 16.90 | 25.10 |
| Pubmed | GCN | 90.20 | 91.20 | 88.70 | 91.30 | 90.10 | 90.90 | 88.10 | 90.30 | 84.60 | 86.20 | 79.70 | 79.00 |
| | ResGCN | - | - | 90.70 | 90.70 | 89.60 | 90.50 | 89.60 | 91.00 | 90.20 | 91.10 | 87.90 | 90.20 |
| | JKNet | - | - | 90.50 | 91.30 | 90.60 | 91.20 | 89.90 | 91.50 | 89.20 | 91.30 | 90.60 | 91.60 |
| | IncepGCN | 90.10 | 90.70 | 89.90 | 91.60 | 90.20 | 91.50 | 90.80 | 91.30 | OOM | 90.50 | OOM | 90.00 |
| | GraphSAGE | - | - | 89.40 | 91.20 | 90.20 | 91.70 | 83.50 | 87.80 | 41.30 | 47.90 | 40.70 | 62.30 |
| Reddit | GCN | 96.11 | 96.13 | 96.62 | 96.71 | 96.17 | 96.48 | 67.11 | 90.54 | 45.55 | 50.51 | - | - |
| | ResGCN | - | - | 96.13 | 96.33 | 96.37 | 96.46 | 96.34 | 96.48 | 93.93 | 94.27 | - | - |
| | JKNet | - | - | 96.54 | 96.75 | 96.82 | 97.02 | OOM | 96.78 | OOM | OOM | - | - |
| | IncepGCN | 96.22 | 96.28 | 96.48 | 96.77 | 96.43 | 96.87 | OOM | OOM | OOM | OOM | - | - |
| | GraphSAGE | - | - | 96.45 | 96.54 | 96.38 | 96.42 | 96.15 | 96.18 | 96.43 | 96.47 | - | - |

