# OpenReview forum: "DropEdge: Towards Deep Graph Convolutional Networks on Node Classification"
_ICLR.cc/2020/Conference — Accept (Poster)_

### Official Review · AnonReviewer2 · 2019-10-23
**Official Blind Review #2**

**Rating:** 3

**Review:**

The authors propose a simple and interesting strategy, DropEdge, to alleviate the over-fitting and over-smoothing in GCN. The logic is simple and clear and the paper is well-written.

Major concerns:
1. After randomly enforce a certain rate of edges to be zero, how to preserve properties in the original complex network, such as degree power-law distribution, communities? If it was not necessary to preserve the properties, then what information should be preserved from the original graph.
2. Randomly drop edges may result in disconnected components, how to handle disconnected components?
3. Why do the authors use dimension difference as the measure to quantitatively evaluate information loss in Thm 1. More dimension reduction does not mean more information loss.
4. As a follow-up concern for C1, graph sparsification makes more sense than DropEdge because it has clear information reserve targets while there is no target for the randomness in DropEdge.
5. In Table 1 and Fig 2, why the improvements for more layers are bigger than those of the fewer layers?
6. In Fig 2, why the trend of Reddit dataset is so different from others (the more layers the more improvements by applying DropEdge)?
7. In Table 2, why there are the DropEdge versions for some methods while not for some other methods (e.g., FastGCN, ASGCN)? Why there is no result of GAT?

Minor:
1. Sec 3, "notation", "\mathbf{x}_n" -> "\mathbf{x}_N"
2. Eq (1), "\mathbf{h}_n^{(l+1)}" -> "\mathbf{x}_N^{(l+1)}"
3. What's C_l in the explaination under Eq(1)?

**Experience Assessment:**

I have published one or two papers in this area.

**Review Assessment: Checking Correctness Of Derivations And Theory:**

I assessed the sensibility of the derivations and theory.

**Review Assessment: Checking Correctness Of Experiments:**

I carefully checked the experiments.

**Review Assessment: Thoroughness In Paper Reading:**

I read the paper at least twice and used my best judgement in assessing the paper.

---

> ### Author Response · Authors · 2019-11-09
> **Response to Reviewer #2**
>
> We appreciate the reviewer for ordering the questions with numbers, which helps us to respond more conveniently.
>
> Q1: DropEdge does change the graph properties for each epoch. But statistically, as discussed in our reply to Q4, Reviewer#4, DropEdge does not change the expectation of neighbor aggregation that plays a crucial role in characterizing input graphs. Hence, the statistics of graph properties are still preserved.
>
> Q2: Drawing for our reply in Q1, DropEdge will not change the connectivity in expectation, even it may result in disconnected components occasionally in one epoch.
>
> Q3: The information measurement in Thm.1 refers to how much freedom we have to describe a point in a certain space. The dimensionality of the space is a natural and direct choice, thus we use dimension reduction to reflect information loss.
>
> Q4: As we discussed in Section 4.3, The purpose of the graph sparsification and DropEdge are different. Graph sparsification aims to remove unnecessary edges of graphs, while keeping almost all information of the input graph, while DropEdge is an efficient approach to reduce the over-smoothing based on our theoretical analysis. Moreover, as mentioned in Q1, DropEdge preserves the statistic of graph properties, and involves no bias.
>
> Q5: According to our theoretical analysis, deeper GNN models suffer from more serious over-smoothing issues than that of shallower ones. It is thus not surprising that DropEdge can gain more improvements from more layers. The experimental results in Tab. 1 and Fig.2 validate our theoretical findings.
>
> Q6: The trend of Reddit dataset is still generally consistent with other datasets if we compare the results of 4/8/32 layers (the more layers the more improvements by applying DropEdge). The corner case happens when the depth is 16. If we check Table 7 in the appendix, there is a huge performance drop in GCN without DropEdge at 16 layers, making the improvement by DropEdge bigger than that of 32 layers.
>
> Q7: The motivation of FastGCN and ASGCN is to speed up GCN, and they can be considered as different efficient implementations of GCN.  We believe performing a comparison on GCN is sufficient without further consideration on FastGCN and ASGCN.  GAT is different from GCN, and we are willing to provide the results below:
> |                                   | Cora  | Citeseer |
> | GAT                          | 0.863 | 0.781     |
> | GAT w/ DropEdge | 0.881 | 0.792     |
> As expected, DropEdge can still enhance its performance.
>
> Minor Q3: $C_l$ refers to the size of $l$-th hidden layer.

---

### Official Review · AnonReviewer3 · 2019-10-30
**Official Blind Review #3**

**Rating:** 3

**Review:**

This paper studied the problem of "deep" GCNs where the goal is to develop training methods that can make GCN becomes deeper while maintaining good test accuracy.  The authors proposed a new method called "DropEdge", where they randomly drop out the edges of the input graphs and demonstrate in experiments that this technique can indeed boost up the testing accuracy of deep GCN compared to other baselines.

This paper is clearly well-written and the authors conducted a comprehensive study on deep GCNs. I also like the discussion in sec 4.3 where the authors explicitly clarify what are the difference between DropEdge, Dropout and DropNode, as the other two are the methods that will pop up during reading this paper. The extensive experiment results also show that for deeper GCNs, DropEdge always win over other baselines (see Tab 1) despite most of them are marginal except the backbone being GraphSAGE on Citeseer. Can you explain why this is the case? Why other backbones seem to have similar performance  even with DropEdge (i.e. most of the accuracy increase are less than 3 %).

Question:
1.  When looking at Tab 1, it looks like most of the time, 2-layers networks are already the best (or close to the best) and are clearly better than 32 layers. Therefore, this makes me wonder: why do we need deeper networks at all if the shallow networks can already achieve a good (almost best) performance and it is also much similar and efficient in training? Can you please clarify why do we care to train a deeper network at all under this scenario? Are there any reasons that we would like to use deeper network as opposed to shallower networks?

2. It is less clear to me regarding this sentence: "DropEdge either retards the convergence speed of over-smoothing or relieves the information loss caused by it"

Overall, I think this paper presents an interesting study on making deeper GCNs comparable to shallow network performance, but since the the boosted performance doesn't really outperform most of the 2-layer networks,  I would like to hear the justification of why we need the deeper networks for this node classification task.

**Experience Assessment:**

I have read many papers in this area.

**Review Assessment: Checking Correctness Of Derivations And Theory:**

I assessed the sensibility of the derivations and theory.

**Review Assessment: Checking Correctness Of Experiments:**

I assessed the sensibility of the experiments.

**Review Assessment: Thoroughness In Paper Reading:**

I read the paper at least twice and used my best judgement in assessing the paper.

---

> ### Author Response · Authors · 2019-11-09
> **Response to Reviewer #3**
>
> We thank the reviewer for accepting the interestingness and the completeness of our paper. We present our responses below.
>
> Q1. About the significance of performance improvement.
>
> As mentioned in Section 5.1, the benchmark datasets are well-studied and well-tuned in the graph learning field. Achieving a 1-2% increase can be regarded as a remarkable improvement.  For the baselines we consider in Tab.1, DropEdge generally improves them around (or bigger than) 1% under different depths, which is significant considering the challenge on these datasets.
>
> Q2. About the claim of our paper on deeper GNNs.
>
> The reviewer possibly misunderstood our claim. Our paper is not showing deeper is better.  Instead, we are more interested in investigating why GCN failed with deep layers and how over-smoothing happened. We hence propose DropEdge, a simple but effective method that is capable of enhancing various kinds of GNNs regardless of the network depth. Our motivation of discussing and reporting the results of varying depth in Tab.1 is to study how much DropEdge can enhance deep GNNs.  The reviewer raised that  "it looks like most of the time, 2-layers networks are already the best (or close to the best)", which is true but only when all models (including the 2-layer ones) have applied DropEdge.
>
> Q3.  About the clarification of "DropEdge either retards the convergence speed of over-smoothing or relieves the information loss caused by it".
>
> This sentence reflects two parts of Theorem 1. As the first part, it retards the convergence speed of over-smoothing and as the second part, it relieves the information loss caused by over-smoothing.

---

### Official Review · AnonReviewer4 · 2019-11-03
**Official Blind Review #4**

**Rating:** 6

**Review:**

The authors propose a simple but effective strategy that aims to alleviate not only overfitting, but also feature degradation (oversmoothing) in deep graph convolutional networks (GCNs).  Inspired by dropout in traditional MLPs and convnets, the authors clearly motivate their contribution in terms of alleviating both overfitting and oversmoothing, which are problems established both in previous literature as well as validated empirically by the authors. Ultimately, the authors provide solid empirical evidence that, while a bit heuristic, their method is effective at alleviating at least partially the issues of overfitting and oversmoothing.

I vote weak-accept in light of convincing empirical results, some theoretical exploration of the method's properties, but limited novelty.

Pros:
Simple, intuitive method
Draws from existing literature relating to dropout-like methods
Little computational overhead
Solid experimental justification
Some theoretical support for the method
Cons:
Method is somewhat heuristic
Mitigates, rather than solves, the issue of oversmoothing
Limited novelty (straightforward extension of dropout to graphs edges)
Unclear why dropping edges is "valid" augmentation

Followup-questions/areas for improving score:

It would be nice to have a principled way of choosing the dropout proportion; 0.8 is chosen somewhat arbitrarily by the authors (presumably because it generally performed well). There is at least a nice interpretation of choosing 0.5 for the dropout proportion in regular dropout (maximum regularization).

As brought up in the comments, edges to drop out to the graph's properties is an interesting direction to explore. While the authors state that they would like to keep the method simple and general, the method is ultimately devised as an adaptation of dropout to graphs, so exploiting graph-specific properties seems reasonable and a potential avenue to further improving performance.

p2: "First, DropEdge can be considered as a data augmentation technique" Why are these augmentations valid; why should the output of the network be invariant to these augmentations? I would like to see some justification for why the proposed random modification of the graph structure is valid; intuitively, it seems like it might make the learning problem impossible in some cases.

Deeper analysis of the (more interesting, I think) layer-independent regime would be nice. (As a side-note, the name "layer-independent" for this regime is a bit confusing, as the edges dropped out *do* depend on the layer here, whereas in the "layer dependent" regime, edges dropped out do *not* depend on the layer).

Comments:
Figure 1 could probably be re-organized to better highlight the comparison between GCNs with and without DropEdge; consolidating the content into 2 figures instead of 4 might be more easily parsable. Adding figure-specific captions and defining the x axis would also be nice.

Use "reduce" in place of "retard"
p2 " With contending the scalability" improve phrasing
p2 "By recent," -> "Recently,"
p2 "difficulty on" -> "difficulty in"
p2 " deep networks lying" -> "deep networks lies"
p3 "which is a generation of the conclusion" improve phrasing
p3 " disconnected between" -> "disconnected from"
p4 "adjacent matrix" -> "adjacency matrix"
p4 "severer " -> "more severe"
p5 "but has no help" -> "but is no help"
p5 "no touch to the adjacency matrix" -> improve phrasing


**Experience Assessment:**

I do not know much about this area.

**Review Assessment: Checking Correctness Of Derivations And Theory:**

I did not assess the derivations or theory.

**Review Assessment: Checking Correctness Of Experiments:**

I assessed the sensibility of the experiments.

**Review Assessment: Thoroughness In Paper Reading:**

I read the paper at least twice and used my best judgement in assessing the paper.

---

> ### Author Response · Authors · 2019-11-09
> **Response to Reviewer #4**
>
> We really thank the reviewer for the recognition of our contributions to the experimental evaluations and the theoretical justification. Here, we would like to provide more explanations to address the reviewer's concerns.
>
> Q1. The novelty of DropEdge.
>
> We agree that our DropEdge is simple and is inspired by Dropout. Yet, when we put it in the context of graph learning, DropEdge is indeed a novel method that is able to alleviate over-smoothing, while Dropout cannot. DropEdge can be regarded as an extension of Dropout to graph edges, but this extension, in certain sense, is not straightforward—people usually adapt the idea of Dropout in GNNs to drop the network activations with less thinking in dropping the network input. Interestingly, simply dropping edges by random is sufficient to deliver promising results, as verified by our paper experimentally and theoretically.
>
> Q2. Choosing the dropout proportion.
>
> The reviewer probably misunderstood the experimental setting. We did not fix the drop proportion $p$ to be 0.8 for the main of our experiments. Instead, as already presented in the second paragraph in Section 5.1, we use the validation set to determine the dropping rate for each benchmark in Tab.1; different datasets could have different dropping rates. In Section 5.2.1 (and Fig.3), we fix it to be 0.8 for a case study on evaluating how DropEdge can prevent from over-smoothing.
> To further illustrate how the dropping proportion actually acts, we have conducted an example experiment for GCN-4 (the best GCN model for Cora in Tab. 2) by varying $p$ from 0 to 1 with an increasing step of 0.2. The results are
> $p$   |  1    | 0.8     | 0.6  | 0.4    |0.2      | 0
> GCN|0.624| 0.778|0.87 |0.869 |0.862 |0.855
> Clearly, it generally improves the performance when $0<p<0.8$. The exceptional cases are $p=0.8，1$ when GCN degenerates (or closely degenerates) to MLP, which is reasonable due to the less expressive power. Furthermore, the best performance is achieved when $0.4 \leq p \leq 0.6$; the selection of dropout proportion $p$ near 0.5 may also be a good choice.
>
> Q3. Exploiting graph-specific properties when considering DropEdge.
>
> Yes, this potentially promotes the performance. Given that our particular interest here is to keep the method simple and general, we are happy to explore more variants of sophisticated DropEdge in future work.
>
> Q4. The justification of why “DropEdge can be considered as a data augmentation technique” is valid.
>
> Thanks for the comment and sorry for the unclear clarification in the current submission. We provide an intuitive understanding here. The key in GNNs is to aggregate neighbors' information for each node, which can be understood as a weighted sum of the neighbor features (the weights are associated with the edges). From the perspective of neighbor aggregation, DropEdge enables a random subset aggregation instead of the full aggregation during GNN training. Statistically, DropEdge only changes the expectation of the neighbor aggregation up to a multiplier $p$, if we drop edges with probability $p$. This multiplier will be actually removed after weights normalization, which is often the case in practice. Therefore, DropEdge does not change the expectation of  neighbor aggregation and is an unbiased data augmentation technique for GNN training. We have added the above specifications in the paper.
>
> Q5. On the layer-independent DropEdge.
>
> Thanks for the comment. Our original purpose of naming it like this is to reflect that DropEdge is conducted independently across layers. If this is a problem, we are willing to rename it as “layer-wise DropEdge” to remove the confusion. We agree that the analysis of “layer-wise DropEdge” is interesting and it is an important future work of theoretical aspects.
>
> Q6. Other comments.
>
> Thanks for the valuable suggestions and we will re-organize Fig.1 accordingly. We will also fix the typos throughout our paper.

---

### Public Comment · ~Alex_Williams1 · 2019-10-02
**Questions with the Experimental Results (Too Good To Be True)**

This results reported in this paper have big problems, especially for the results on Pubmed dataset. It is suggested for the authors to release the source code for the community to check the results. Their results are too good to be true.

(1) Inconsistent with GCN raw model

According to the Table 1 in the paper, the reported results for GCN (i.e., original GCN) are inconsistent with the original GCN paper. For instance, in [1], the accuracy of GCN on Citeseer, Cora, and Pubmed are
70.3% 81.5% 79.0%

However, according to this submission, in Table, the accuracy rate of the original GCN on these three datasets are
79.34% 86.64% 90.22%

These results are all inconsistent and have big problems, especially for the results on Pubmed dataset. If the authors really have run the models on Pubmed dataset, it is extremely hard to achieve an accuracy score higher than 80%. Compared with all the works on the Pubmed datasets according to [2] and several other datasets [4], the scores reported in this paper are all over-exaggerated. The authors probably may need to clarify how they get 90.22% for GCN on Pubmed and so high scores on the other datasets.

(2) Inconsistent with their arXiv version

The authors also release a version at arXiv (submitted on Sept 9, 2019), and the results reported in this paper are also inconsistent with their arXiv version [3]. All the scores in this paper are much higher than their latest arXiv version. The authors may also need to clarify this.

(3) Surprisingly GOOD performance

What's more, for their own method, their accuracy on Pubmed is 91.7%. To be honest, the results are too good to be true. I cannot really trust this scores. The authors do need to clarify how they get such scores and also release the source code out for the community to check the reported results.


[1] SEMI-SUPERVISED CLASSIFICATION WITH GRAPH CONVOLUTIONAL NETWORKS
[2] https://paperswithcode.com/sota/node-classification-on-pubmed
[3] https://arxiv.org/pdf/1907.10903.pdf
[4] https://paperswithcode.com/task/node-classification

---

> ### Public Comment · ~Not_Alex_Williams1 · 2019-10-03
> **Sorry, what? You want us to not trust these results based on comments from your fake id?**
>
> Hi Alex Williams,
>
> I ran a quick check through the MIT directory and found no person named "Alex Williams". Could you please update your institute and other personal information to be more accurate before you troll papers? This isn't the only paper, I've seen innumerable trolls with fake accounts.
>
> I am not sure if the organizers did us authors a favor by disabling anon comments -- now people just make fake ids like this, and troll people.

---

> > ### Public Comment · ~Alex_Williams1 · 2019-10-03
> > **Academic open discussion, no swearing words please, looking forward to your source code**
> >
> > Not sure if I should reply to your rude and mean reply or not. To be polite, I think I still want to.
> >
> > (1) I was a research scientist at MIT. You don't know me, it is fine but it doesn't mean my comments and questions on your paper are not important. It is really a shame for you to do this kind of attack instead of defending your paper with solid results and with your source code.
> >
> > (2) I'm discussing about the problems with this paper, so please respond to my academic questions directly. If you think your paper has no problem, please release your source code. Otherwise, I don't think the community can trust the results reported in your paper.
> >
> > (3) Please don't use swearing words like "shit post" anymore, it is a shame to do such personal attack in your response. Everyone in the community can see your post and your response.
> >
> > (4) Looking forward to seeing your code, you can send the link of your code in the response to this post. I think the community will be interested and happy to check it together. The datasets are benckmark datasets, you don't need to share them and we can download it from the web to ensure you didn't change the data by yourself.
> >
> > PS. We all want to see how you achieve 91.7% accuracy on Pubmed. Frankly speaking, if your method can do achieve such a big improvement, the community and me will all be happy to support your work.
> >
> > Also if you are not the author of this paper, please show YOUR NAME and YOUR AFFILIATION. If you are the author of this paper, then it is a shame for you to create this non-existing account to defend your paper. It is so ridiculous and not funny at all... You are not very serious about your results, your submission, your response and your manner.

---

> > > ### Public Comment · ~Not_Alex_Williams1 · 2019-10-03
> > > **Not an author of this paper!**
> > >
> > > Sorry about the "swear" word. I have nothing to do with this paper, and it is quite funny how you assume that the authors would actually go to such lengths to do this. The authors posting would display as an "Official paper comment" or something along those lines?
> > >
> > > PS. I am not being critical of your comment, but rather of the fact that your feedback is not constructive and you choose to be extremely critical of a work under the veil of a fake account. Please show us YOUR REAL NAME, and YOUR REAL AFFILIATION, and maybe also include a link to your website and some of your own papers in your profile.
> > >
> > > PPS. If I remember correctly, you profile stated you were still at MIT earlier and now it says you left at 2016. I think the organizers had an agenda in mind when they stopped anonymous comments -- to stop anonymous comments! Commenting from a fake-id doesn't really help the organizers accomplish what they intended to.

---

> > > > ### Public Comment · ~Alex_Williams1 · 2019-10-03
> > > > **Thanks for your participation as a non-author of this paper**
> > > >
> > > > (1) If you also work on GNNs, you will understand what I mean in my comments. Node classification on Pubmed is extremely hard, and 80% accuracy is already very hard to achieve. In this paper, the authors achieve 91.7% accuracy. Don't you feel it is interesting and wanna take a look into their model source code?
> > > >
> > > > (2) I don't want to impeach this paper by mistake, so I suggest the authors to release the source code out and the community can check their performance. Is it wrong? What do you mean by "constructive"? They publish a paper with ridiculously high and inconsistent scores, and we want to check their correctness, isn't it constructive in your mind?
> > > >
> > > > (3) Alex Williams is my true name and I also show my real workplace. Please stop the personal attack. Try to defend the paper by showing me my questions are wrong and the scores in this paper are correct. By the way, may I kindly ask  "What is your name, Mr. Not Alex Williams ?" You haven't taken your mask off yet. If you are not the author, "How can you respond to my comments to this paper so quickly without system notifications?"
> > > >
> > > > (4) Anyway, let's focus on this paper. Really don't wanna continue the meaningless squabbles with you like kids

---

> > > > > ### Public Comment · ~Not_Alex_Williams1 · 2019-10-03
> > > > > **Biasing the reviewers?**
> > > > >
> > > > > Academic Curiosity/Constructive Feedback: You could pose it as a question and mention that you've noticed different accuracies, and ask the authors if they have a reasonable explanation for this. I could draft an example response for you.
> > > > >
> > > > > As a mature researcher, I am sure you understand that tone matters, and having reviewed in multiple venues, it is often an instruction to maintain a positive tone and not be condescending. If the primary author on this is a early graduate student, even if he accidentally evaluated his numbers on a wrong dataset or on the train set -- attacks such as yours will make him question his desire to participate in science. You could also politely ask for source code as opposed to being vicious in your criticism.
> > > > >
> > > > > Words such as "Untrustworthy Experimental Results" come across as an attempt to impeach the paper/Bias Reviewers.
> > > > >
> > > > > PS. The internet doesn't guide me to a Mr. Alex Williams who was a research scientist at MIT. Can you share your webpage/work email, so we can continue this discussion offline?

---

> > > > > > ### Public Comment · ~Not_Alex_Williams1 · 2019-10-03
> > > > > > **Notifications**
> > > > > >
> > > > > > I get a notification whenever anyone responds to my comments. As simple as that :)

---

> > > > > > ### Public Comment · ~Alex_Williams1 · 2019-10-03
> > > > > > **Huh, double standard. Why you delete your curse words ? Looking forward to seeing the source code**
> > > > > >
> > > > > > If the authors can release the code and clarify my concerns in the very beginning, I will be very happy to revise and even delete my post. The authors didn't show up, but you come as a trouble maker. Be serious and don't make things worse... If you are the author of this paper, you already mess things up.
> > > > > >
> > > > > > I'm sure you know what you are doing, please show your respect to this paper and my questions. I like and also respect this paper, so I take the time to clarify my concerns over again to the authors and to YOU. Really looking forward to seeing the source code.
> > > > > >
> > > > > > PS. I see you delete your curse words, and then you come back to impeach me about my tone with your double standard ?... Huh, you are so funny. There is no respect for you (a coward behind mask) anymore, this will be my last post...
> > > > > >
> > > > > > PSS. Will not change my post as you may wish. I don't know who you are, but you also need to be responsible for your posted curse words (even though you delete them), rudeness and your double standard.

---

> > > > > > > ### Public Comment · ~Not_Alex_Williams1 · 2019-10-03
> > > > > > > **Please share your webpage/official email, and I will reach out to your personally and discuss this offline.**
> > > > > > >
> > > > > > > I changed it because apparently the word "shit" is a swear word, and offensive -- I recommend you read some of Jonathan Swift's works. I changed the word because it hurt someone's sentiments, I have no expectation that you edit/modify your comment.
> > > > > > >
> > > > > > > And, for the sake of the authors, I'd like to clarify that I am in no way affiliated with the authors. This is a fake id, just like yours. I will end this here, and not disrespect the author's efforts anymore. I hope the community takes notice of your attempts at disgracefully attacking this paper with malicious intent.
> > > > > > >
> > > > > > > PS. That aside, I think it's great to have authors release source code but I think there's better ways to asking for the same.
> > > > > > >
> > > > > > > PPS. Why don't you post a link to your webpage, and I'll take off my "cowardly mask"?

---

> > > > ### Public Comment · ~Alex_Williams1 · 2019-10-03
> > > > **Looking forward to seeing the source code**
> > > >
> > > > PS. You make things more interesting now. I'm much more eager to see the source code of this paper after reading your response.
> > > >
> > > > I didn't expect the fast response from you actually. You can reply faster than the authors, huh... Hope the authors will thank you for your great “help”.

---

> ### Author Response · Authors · 2019-10-04
> **The performance difference is due to the different experimental setting**
>
> Hi, Alex Williams.
>
> Thanks for your interest in our work, and sorry for the delay response.
>
> (1)	Please note that the training-testing division of the datasets in this paper is different from that in the original GCN paper. As already mentioned in the last sentence of the first paragraph in Section 5, we follow the setting in FastGCN [1] and AS-GCN [2] to use full supervised data for the training (while the original GCN use the semi-supervised setting). This is why we obtained higher numbers for the same GCN model on Cora, Citeseer, and Pubmed. Our performance with DropEdge reaches 91.7% compared to 90.22% by GCN on Pubmed under the full supervision setting, which is still reasonable. The main purpose of this paper is to demonstrate the impact of DropEdge on promoting deep GCNs (see Table 1 and Table 7) without particular concern on what setting we prefer.
>
> (2)	Our code is available in https://github.com/DropEdge/DropEdge . Feel free to check. If you have any question on running it, please tell us.
>
> [1] Jie Chen, Tengfei Ma, and Cao Xiao. Fastgcn: Fast learning with graph convolutional networks via importance sampling. In Proceedings of the 6th International Conference on Learning Representations, 2018.
> [2] Wenbing Huang, Tong Zhang, Yu Rong, and Junzhou Huang. Adaptive sampling towards fast graph representation learning. In Advances in Neural Information Processing Systems, pp. 4558–4567, 2018.

---

### Public Comment · ~William_H_Cohen1 · 2019-10-04
**Request for code!**

Dear Respective Authors,
If it is possible, could you please release the code to verify the results? When the code could be released? It would be great to have some response from you.
Best regards, WCohen

---

> ### Author Response · Authors · 2019-10-04
> **The link of our source code**
>
> Hi William H Cohen,
>
> Thanks for your interest in our work. Our code can be downloaded from
> https://github.com/DropEdge/DropEdge

---

> > ### Public Comment · ~Huaxin_Song1 · 2019-10-07
> > **Link not available?**
> >
> > As titled. Thanks!

---

> > > ### Author Response · Authors · 2019-10-07
> > > **Please remove the last dot**
> > >
> > > Hello Huaxin,
> > > The link wrongly includes the last dot. Please remove it to reach the correct website. The correct link is
> > >
> > > https://github.com/DropEdge/DropEdge
> > >
> > > Best,

---

### Public Comment · ~Alex_Williams1 · 2019-10-04
**It's unfair. Everyone is racing to compete, you cannot take a rocket to increase your scores via changing the train/val/test ratios by yourself... I will keep fighting against your misconduct in research forever!**

(1) Firstly of all, thanks for showing up finally and providing your source code.


(2) Can you please also provide the results on regular semi-supervised setting (like [1], GCN, GAT)?

Cora, Citesser and PubMed are all benchmark datasets, you cannot change the train/test ratio as you may want just to increase your scores. It is not serious research any more...

Not clear about your model based on the normal settings, so we can see the TRUE improvement.


(3) Everyone is racing together to compete, you cannot take a rocket by breaking the rules only to increase your scores.

Don't you think it is unfair for the other published and existing papers ? You break the record and get Number One. So what? You cheat and indiscriminately change the settings.

It doesn't make sense at all by using the one or two bad example papers to demonstrate your motivation is correct and reasonable. One of the papers [2] listed in your response is also from you authors, as the rule breaker. You break the rule before, and use your misconduct as an evidence to support you to break the rules again?

You get 91.7% on Pubmed by cheating and changing the train/val/test ratios ! Have you thought about the existing papers and researchers, who obey the rules and get accuracy below 80%, like [3]?  How will they survive in the competition with you? I will keep fighting against such misconduct in research with you forever.


(4) According to Table 2, Your results only show your methods can improve GCN and GraphSage. How about the others you use in Table 2 ? Do you think if you can provide the results of the missing vanilla methods as follows?
ResGCN
JKNet
IncepGCN

as well as the results of your boosted DropEdge versions of several methods as follows?
FastGCN + EdgeDrop
ASGCN + EdgeDrop

It will make the comparison more complete.


(5) Will check your source code and let you know if I have any problems in reproducing all the reported results.


[1] Prithviraj Sen, Galileo Namata, Mustafa Bilgic, Lise Getoor, Brian Galligher, and Tina Eliassi-Rad.
Collective classification in network data. AI magazine, 29(3):93, 2008.
[2] Adaptive Sampling Towards Fast Graph Representation Learning
[3] AdaGCN: Adaboosting Graph Convolutional Networks into Deep Models

---

> ### Author Response · Authors · 2019-10-08
> **We only accept constructive and valuable discussions but not offensive and impolite comments.**
>
>
> (1)	First of all, we believe that the ICLR open review here is a platform that only accepts constructive and valuable discussions on each submitted paper. If you respect this and are really interested in our paper, please remove your offensive words like “cheat”, “indiscriminately change the settings”, “misconduct”, “keep fighting”, “the results are too good to be true”.
>
> (2)	Your accusations on our experimental setting are rude and not reasonable by any means.
>
> (2.1) First, we have stated our settings clearly in the beginning of our experiments. We have no way to “cheat” anyone. The full supervised setting is originally introduced by FastGCN (not AS-GCN ). Our using the same setting as FastGCN here is due to our concern that both Cora and Citeseer are too small for benchmarking and it could incur bias if we keep using part of the labelled data.
>
> (2.2) Second, all compared methods in our experiments are conducted in the same setting. It is an apple-to-apple comparison. We have never contrasted our numbers against those semi-supervised methods, and of course we are never meant to. Your comment on saying our comparisons are unfair is unfair itself.
>
> (2.3) Different from the other three datasets, the Reddit dataset proposed by GraphSAGE is used under full supervision, which is consistent with our paper.
>
> (2.3) Finally, research is not just about competition. We believe that professional researchers in the community will respect a paper for its novelty, technicality and interestingness, not just because it can beat all methods under the machine-parsable setting.
>
> (3)	Still, we are willing to provide the results under the semi-supervised setting on Cora, Citeseer and Pubmed. We obtained the following results on 2-layer GCN:
>
> | orginal | no-dropedge (ours) | dropedge (ours) |
> --------------------------------------------------------------------------------
> Cora   | 81.5 | 81.1 | 82.8 |
> Citeseer | 70.3 | 70.8 | 72.3 |
> Pubmed | 79.0 | 79.0 | 79.6 |
>
> The results in the first column are from the original GCN paper, and those of the last two columns correspond to the GCNs w/o and w DropEdge, respectively. Note that the results w/o DropEdge are comparable to those in the GCN paper, demonstrating the reliability of our experiments; adding DropEdge consistently promotes the performance on all three datasets. We will add more results on other backbones if necessary.
>
> Overall, we sincerely hope you to show more respect to our work, and continue an encouraging discussion on the motivation, formulation and interestingness of our method. If you keep using offensive tone, we will refuse to respond to any question by you.

---

> > ### Public Comment · ~Alex_Williams1 · 2019-10-08
> > **Give up your dirty trick. Show your respect for other researchers' efforts, then you will receive respect from them as a reward.**
> >
> > Based on your provided results, do you still think your methods can outperform the existing approaches [1]?
> >
> > Thanks again for providing your results based on the regular semisupervised setting. Please don't mis-interpret my words. I don't care about competition, I only care about fairness.
> >
> > I can also write a paper, and increase the scores to more than 99% by changing the train/test ratios as what you do. Do you think such papers make sense or not for you? It is also not healthy for the community development, if everyone acts as dirty as your team.
> >
> > Not sure who in your author team decides to play "dirty". It really makes other researchers feel sad and frustrated. If the bad idea is from the young people, it is still suggested to show your respect for the other researchers' efforts, since you still have a long academic journey. Give up your "dirty trick" and you will receive the respect from the community as well. This is the "constructive" suggestion from me.
> >
> > If you still insist and argue your dirty tricks are correct, I will keep fighting against such misconducts with you. I don't really know you authors nor your institute, but your team has a really bad impression for me.
> >
> > PS. This is my last post. We will stop here.
> >
> >
> >
> > [1] https://paperswithcode.com/task/node-classification

---

> > > ### Comment · Area_Chair1 · 2019-10-10
> > > **Please stop**
> > >
> > > I would appreciate if you stop offending the authors and stick to academic matters in all further discussions including this one.
> > >
> > > Having different splits or settings for the same dataset is not a "dirty" trick, as long as the split is clearly specified, especially given that the authors have followed several previous publications that had used the same splits.

---

> > > > ### Public Comment · ~Alex_Williams1 · 2019-10-10
> > > > **Cannot agree with your point of view , but will stop as suggested.**
> > > >
> > > >
> > > >
> > > > Cannot agree with you at all , but will stop as suggested.

---

### Public Comment · ~Deli_Chen1 · 2019-10-09
**Not All Edges Should be Removed**

Thank you for your work.

(1) The proposed DropEdge randomly removes all edges from the input graph, but in our recent study[1], we have proven that it is the intra-class edges that make GNN models work on the node classification task, which play an important role and should not be removed.  We also prove[1] that the over-smoothing issue is caused by the over-mixing of information and noise, which is partly caused by the inter-class edges. So it is the inter-class edges instead of all edges that should be randomly removed.

(2)  [2] performed a theoretical analysis on GCN, and conclude that performing smoothing operation on node representations is the key mechanism why GCN work. [1] proposed that smoothing is inevitable for various GNN models. So we think the measurement for over-smoothing should pay more attention to the difference of inter-class nodes' representations instead of all nodes' representations. A solution is also given in [1].

[1]Deli Chen, Yankai Lin, Wei Li, Peng Li, JieZhou, Xu Sun:  Measuring and Relieving the Over-smoothing Problem for Graph Neural Networks from the Topological View (https://arxiv.org/abs/1909.03211)
[2]Li, Q.; Han, Z.; and Wu, X.-M. 2018. Deeper Insights into Graph Convolutional Networks for Semi-supervised Learning.

---

> ### Author Response · Authors · 2019-10-12
> **Dropping Edges by random is simple yet effective, which can prevent overfitting as well.**
>
> We appreciate your interests in our work. Your raising paper [1] brings us a different idea to measure over-smoothing and the approaches to relieve it. It is an interesting work. Here, we would like to take it as a chance to discuss the difference between [1] and our paper:
>
> (1)	The different understanding on over-smoothing. While [1] considers over-smoothing as the issue that node representations become identical and indistinguishable as network depth increases, this paper follows previous studies [2,3] that prefer to define it as the convergence of node representations to a stationary distribution (or a subspace as proved by [3]). The latter understanding admits that the convergent representation of different node could be different but it is only topology-aware and independent to the initial input. This is consistent to the random walk theory. Guided by this, we proposed DropEdge, a novel method to slow down the speed of convergence; as shown by Theorem 1, we have proved that dropping any edge (not just the inter-class ones) is able to retard the speed of over-smoothing or relieve the information loss caused by it.
>
> (2)	DropEdge is simple yet effective. We agree that it could become more sophisticated if we apply certain heuristics other than randomness to determine which edge to be deleted (e.g. the inter-class edges). However, it will inevitably involve more complexity, and is impractical when the node labels are unknown (such as unsupervised learning). By contrary, DropEdge is efficient and applicable for broader cases. More importantly, as shown by our experiments, dropping edges by random is sufficient to enhance the performance of a variety of both shallow and deep GCNs.
>
> (3)	DropEdge can prevent over-fitting as well; in this sense, it is more like Dropout. As already presented in our paper, DropEdge can be considered as a data augmentation technique. By DropEdge, we are actually generating different random deformed copies of the original graph; as such, we augment the randomness and the diversity of the input data, thus better capable of preventing over-fitting.
>
> [1] Deli Chen, Yankai Lin, Wei Li, Peng Li, JieZhou, Xu Sun: Measuring and Relieving the Over-smoothing Problem for Graph Neural Networks from the Topological View.
> [2] Johannes Klicpera, Aleksandar Bojchevski, Stephan Günnemann: Predict then Propagate: Graph Neural Networks meet Personalized PageRank.
> [3] Kenta Oono, Taiji Suzuki: Graph Neural Networks Exponentially Lose Expressive Power for Node Classification

---

### Author Response · Authors · 2019-10-14
**The results of semi-supervised node classification.**

Hi All,
      We updated our code to support the semi-supervised setting of node classification. All semi-supervised classification results of Cora, Citeseer and Pubmed are available in the GitHub.

      Please check out them if you are interested in our work.
      Link: https://github.com/DropEdge/DropEdge

Thanks!
Authors

---

### Public Comment · ~Petar_Veličković1 · 2020-01-30
**Relationship to prior regularisers**

Very interesting work! Thank you for such a rigorous evaluation and the theoretical justification.

From how I understood the method, it is very similar to the regularisation we employed in the GAT paper (https://openreview.net/forum?id=rJXMpikCZ ); as per Section 3.3:

"Furthermore, dropout (Srivastava et al., 2014) with p = 0.6 is applied to
both layers’ inputs, *as well as to the normalized attention coefficients (critically, this means that at
each training iteration, each node is exposed to a stochastically sampled neighborhood)*."

Performing dropout on the attention coefficients should be more-or-less along the same lines as edge dropping? If so, perhaps this link should be better highlighted, and in relation to the "GAT w/ DropEdge" baseline you mentioned in a rebuttal comment below, does this mean that the result for regular GAT was obtained without this dropout?

Thank you!

---

> ### Author Response · Authors · 2020-02-09
> **More clarifications**
>
> Hi, Petar. Really appreciate your interest in our work.
>
> Sorry for missing the part you mentioned in your paper. Yes, performing dropout on the attention coefficients should be a specific form of GAT with DropEdge, and the result for regular GAT was obtained without this dropout.
>
> Having said that, there are several different points inbetween. First, your version is indeed a post-conducted version of DropEdge, as you compute all attentions prior to attention dropout. Here, in our work, we first perform DropEdge and then use GAT, avoiding unnecessary computation of edge attentions. Besides, we further perform adjacency normalization following DropEdge, which, even simple, is able to make it much easier to converge during training. Without normalization, it will also intensify gradient vanish as the number of layers grows.
>
> More or less, the attention dropout seems an ad-hoc trick in your paper, and the relation to over-smoothing is never explored. In our paper, however, we have formally presented the formulation of DropEdge and provided rigorous theoretical justification of its benefit in alleviating over-smoothing. We also carried out extensive experiments by imposing DropEdge on several popular backbones.
>
> We are happy you raising this discussion, and have added the above connection in the final copy.

---

### Public Comment · ~Dongsheng_Luo1 · 2020-02-26
**Some questions about theoretical analysis.**

Dear authors,

I think this is a very interesting paper.  I have some concerns about the theoretical analysis.

1) Formulation of l*.  According to Definition 3, l* is the minimal value of the layers that satisfy Equation 3 and formulation is given in Appendix Lemma 2, Equation (6).  since l* is the minimal, I think dM(H(l-1)) >=epsilon should be proved.


2)  It seems that Definition 4 of conductance is not the standard form. According to the wiki and your reference (László 1993),   the denominator should be min(V(S), V(\bar(S)), where V(S) is the sum of node degrees in S.

3) If we adopt the standard form conductance, the statement "the conductance of the graph can only decrease if one edge is removed from the graph" may not hold.  Intuitively, we remove an edge inside a cluster, then the conductance of the graph increases.

Here is a toy example: https://github.com/flyingdoog/DropEdge-tf/blob/master/DropEdge.ipynb

V = [0,1,2,3,4,5]
adj_list = {}
adj_list[0]=[1,2,3]
adj_list[1]=[0,2]
adj_list[2]=[0,1]
adj_list[3]=[0,4,5]
adj_list[4]=[3,5]
adj_list[5]=[3,4]

The conductance is 0.1429, removing the edge(4,5) leading to 0.2.

Thank you!


[1] László Lovász et al. Random walks on graphs: A survey. Combinatorics, Paul erdos is eighty, 2(1):
1–46, 1993

---

> ### Author Response · Authors · 2020-03-08
> **More Clarifications**
>
> Hi, Dongsheng,
>
> Really appreciate your interest and the questions your raised for our paper. Based on your comments, we recognize that there are indeed unclear and imprecise descriptions in our current version. But note that the main story of our paper still holds. We summarize our clarifications below.
>
> 1)	Our derivation is based on the work [1] where the over-smoothing is characterized in an asymptotical form, so we can only justify the increment of the RELAXED smoothing layer (which is an upper bound of the true smoothing layer) in Theorem 1. We have extra defined the relaxed smoothing layer in our paper and reflected the revisions in Theorem 1 and the proofs in the appendix.
>
> 2)	Your illustration on the conductance is right. Our original proof is by connecting the (relaxed) smoothing layer with the conductance. But, we later find that using conductance is unnecessary. Instead, the proof is better explained via the connection with the resistance (see Eq.(6) in the new appendix). We have rearranged the proof and rigorously shown that the resistance (thus the relaxed smoothing layer) will increase if sufficiently edges are dropped.
>
> Thanks for your questions, which make our paper more rigorous and well-qualified. We hope our explanations help.
>
> [1] Kenta Oono, Taiji Suzuki: Graph Neural Networks Exponentially Lose Expressive Power for Node Classification

---

### Decision · Program_Chairs · 2019-12-19

**Decision:**

Accept (Poster)

**Comment:**

The paper proposes a very simple but thoroughly evaluated and investigated idea for improving generalization in GCNs. Though the reviews are mixed, and in the post-rebuttal discussion the two negative reviewers stuck to their ratings, the area chair feels that there are no strong grounds for rejection in the negative reviews. Accept.